# Influence of Positive Thinking Ideology on Physician Representations of Cancer

**DOI:** 10.3390/bs13100866

**Published:** 2023-10-23

**Authors:** Luis Felipe Higuita-Gutiérrez, Diego Alejandro Estrada-Mesa, Walter Alfredo Salas-Zapata, Jaiberth Antonio Cardona-Arias

**Affiliations:** 1Facultad de Medicina, Universidad Cooperativa de Colombia, Medellín 050016, Colombia; diego.estrada@campusucc.edu.co; 2School of Microbiology, Universidad de Antioquia, Medellín 050016, Colombia; walter.salas@udea.edu.co (W.A.S.-Z.); jaiberth.cardona@udea.edu.co (J.A.C.-A.)

**Keywords:** cancer, the ideology of positive thinking, qualitative research, positive psychology

## Abstract

To understand the influence of positive thinking ideology on cancer representations among physicians in the city of Medellín. Methods: This qualitative study was conducted on the basis of the theoretical and methodological elements of Corbin and Strauss’s grounded theory. Fourteen physicians were included and selected according to the criteria of maximum variation for education, years of study, and personal and family history of cancer. The information was collected through semi-structured interviews and analyzed with open, axial, and selective coding. Results: the ideology of positive thinking has managed to permeate the medical discourse and the representations that they form about the etiology and treatment of cancer. Physicians place the mind, emotions, attitude, and positive thinking as determinants of the origin of the disease and the response to therapy. To argue this link, they use two strategies: (i) a sophisticated and specialized discourse that involves relationships among thoughts, genetics, the neurological, immune and endocrine system and (ii) a mystical and less rational discourse that emphasizes the omnipotence of the mind and thoughts. In no case was the idea of positive thinking rejected or in disagreement with this style of thinking expressed. Conclusion: The fact of linking the disease with mental factors refers to the mind–body dualism and generates a responsibility of the patients on the etiology and therapeutics of the disease, as well as an erasure of the social and political determinants of cancer. The technical discourse and the symbolic capital of physicians offer scientific legitimacy to these ideas and can become performative for patients.

## 1. Introduction

A health insurance company in Colombia, with approximately two million affiliates, published an article on its “Camila: when positive attitude overcomes cancer”. The article tells the story of a 7-year-old girl diagnosed with an osteosarcoma that affected 80% of her body; therefore, she had little chance of survival. Despite this prognosis, the girl maintained a charming personality, always trying to smile when leaving chemotherapy, and it was said that she sang during her hospitalizations. The author stated that the doctors believed Camila’s attitude contributed greatly to her recovery [1]. In the same vein, one of the most widely read newspapers in Cali, the third most important city in the country, published a text titled “The powerful effect of a positive attitude on curing of diseases”. The text begins with the following phrase: “The good attitude of a person influences the process of overcoming of an adverse situation, such as being diagnosed with a disease such as cancer or COVID-19” [2].

The assumption that a positive attitude improves the chances of survival of cancer patients has its empirical basis in a study published in 1979, which followed 69 women with breast cancer for five years. The results revealed that patients who faced the disease with optimism and determination to fight against cancer were more likely to survive than those who faced it with feelings of frustration and hopelessness. Subsequent studies have disproved this finding [3], but the idea persisted. Currently, in the field of psycho-oncology research, it is common to find the belief that cancer survival is related to concepts such as fighting spirit, positive attitude, and certain personality and coping styles [4,5,6].

However, this idea is not exclusive to the health field. In fact, authors such as Barbara Ehrenreich [7], Edgar Cabanas, and Eva Illouz [8] argue that this is the product of the ideology of positive thinking that appeared in the 19th century in American culture, became popularized in the 20th century, and turned ubiquitous in the 21st century, present in countries such as China, the United Arab Emirates, and India, as well as the financial, educational, and military sectors. This ideology is based on the belief that thoughts can modify the physical world and act as a magnet that attracts and materializes what is thought. People should exercise themselves in thinking positively (experiencing positive emotions, being optimistic, seeking happiness) and avoid or repress negative thoughts; otherwise, they would be solely responsible for their unhappiness, ruin or illness.

The diffusion of this ideology has occurred through an assembly of dispositives that include educational institutions, media, managerial literature aimed at the executive class, self-help literature aimed at the public, and coaching. The lexicon, techniques, and scientific legitimacy on which it is based are found in positive psychology [7,8].

Returning to cancer, in the academic literature, cancer is defined as a proliferation of cells that reproduce and divide in an uncontrolled and abnormal manner compared to normal cells. There are various types of cancer classified based on their organ or tissue of origin, as well as the molecular characteristics of cancer cells. Cancer encompasses a heterogeneous group of diseases; however, they share common characteristics that warrant a grouped analysis. This field is particularly attractive for the diffusion of this ideology for several reasons: (i) the belief that thoughts can modify the material world implies that the origin and treatment of cancer are found in them; therefore, it offers patients something to do against a disease for which effective therapeutic alternatives are scarce and against which they usually feel powerless; (ii) doctors are stripped of the monopoly of caring for cancer patients and a field of opportunities is created for the intervention of coaches, the self-help industry and other therapists of the same style; (iii) it is estimated that one in five people in the world will develop cancer during their lifetime [9] and, if families are added to this, a market niche is formed that involves millions of dollars. For example, a search in the self-help section of Amazon yields more than 10,000 books related to cancer. A title that illustrates very well what was narrated in the previous paragraphs is the following: “Cancer Patient Affirmations: Positive Daily Affirmations to Help You Heal Cancer Naturally and Cope with the Emotional Distress Using the Law of Attraction, Self-Hypnosis, Guided Meditation” [10].

Previous studies have shown that patients with cancer embrace the principles of this ideology and relate them to the etiology and treatment of their disease. A study in American cancer patients presented the following testimony to illustrate the patients’ beliefs about the effect of positive thoughts and attitudes on their treatment: “I want to feel confident, because you hear people say, well, attitude is a lot … has a lot to do with your, you know, with, with your health, and you know, fighting chance, especially” [11]. In nurses from an oncology service, it has been found that there exists a belief that a positive attitude helps patients to fight the disease and respond better to treatment [5]. In physicians, there are no studies on the influence of this ideology on the representations of the etiology and treatment of cancer.

The study of this phenomenon in physicians is important because in a debilitating and potentially fatal disease like this patients face problems that do not have clear medical solutions. They are presented with therapeutic alternatives that can be difficult to assess rationally, and to this are added moral dilemmas and social difficulties [12]. Therefore, patients prefer (even are forced to) transfer to physicians a good part of the decision making related to their health and their daily life [13].

Physicians could have beliefs similar to those described in patients or nurses, and this style of thinking generates an additional burden for patients, secondary to their illness, in that they are forced to feel a certain way and to annul feelings of the stages of grief that occur after a cancer diagnosis [14,15]. It has also been described that when the disease progresses, patients tend to blame themselves for their inability to influence the progression of cancer, feel shame and despair [15], a problem that could be exacerbated if treating physicians are also aligned with the positive thinking ideology. In addition, doctors could influence patients to abandon conventional treatments and choose pseudo-therapies related to positive thinking [15]. Therefore, this study was designed to understand the influence of positive thinking ideology on cancer representations among physicians in the city of Medellín.

## 2. Methods

### 2.1. Type of Study

A grounded theory was developed, understood as a set of techniques and procedures that allow the identification of concepts and their relationships from qualitative data [16]. Given the nature of the object identified and analyzed in this study, which pertains to the ideology of positive thinking in representations of cancer, it is considered that the tools and procedures provided by this approach are the most suitable. This is due to the following reasons. First, because the identified and analyzed object is related to beliefs, images, and actions that determine a way of understanding and responding to a problematic event, such as representations of the illness, not the illness itself, but rather the perspectives or ways of assuming and understanding the social reality of agents such as doctors in relation to their professional roles. Second, because it is an “interactive object”, meaning an object that mutates and varies depending on the appropriation and use by agents. In the present work, there is an interest in understanding these appropriations and how they operate in specific understandings within the selected audience. In this regard, it is worth recalling the epistemological foundations of symbolic interactionism, which suggest (1) that people act based on meanings; (2) that these meanings are derived from social interaction; (3) and that they are modified through the interpretive processes of agents in their social interaction.

### 2.2. Selection of Participants

Physicians with training and work experience in the city of Medellín, Colombia were included. The selection of participants was strategically performed to gather the greatest diversity of experiences and opinions regarding the etiology and treatment of cancer. For this purpose, physicians with wide differences in age, years of experience as physicians, clinical specialty, gender, religious beliefs and family or personal history of cancer were selected. The selection was performed one by one and the information was analyzed simultaneously, which allowed deciding the characteristics that the next physician to be chosen should meet in such a way that saturation of the categories was ensured [17]. Information saturation was achieved with 14 physicians and 25 h of recording. At this point, sampling was stopped because the data became redundant and nothing new emerged.

### 2.3. Interviews

Semi-structured interviews were conducted and each physician was interviewed between two and three times according to the stages of grounded theory. The first interview had a general outline that began by inquiring about some aspects of the physicians’ life history and then about the meanings of cancer, its causes, and treatment. The deepening of these general topics was achieved with spontaneous questions from the interviewers that varied between individuals and depended on the flow of the conversation. In the second and third interviews, the development of the concepts identified in the first interview was initiated, clarifications were requested and categories that had an incipient development were complemented.

Semi-structured interviews have become a valuable tool for accessing and identifying the ideological underpinnings of many of the cancer representations within the studied population. On the one hand, the probing nature of this instrument allows us understanding that beneath the surface of initial statements and responses, there exist other questions that can expand upon the information provided by the individuals involved. On the other hand, the flexible format of this instrument allows participants expression of their viewpoints in a more spontaneous manner, a situation that might not be achievable with standardized questionnaires or interviews.

In these meetings, it was also verified that the researchers’ interpretations coincided with what the physicians expressed. All interviews were recorded in an audio format and transcribed in their entirety.

### 2.4. Information Analysis

First, immersion in the data was performed by listening to the interviews repeatedly and reading the transcriptions. The interviews were segmented into phrases or paragraphs and compared with each other, in order to look for fragments that expressed a shared idea. These fragments were grouped with a code or concept that identified them. In the next stage, categories (abstract concepts with greater explanatory power) and subcategories were identified, and relationships between them were established. Finally, a central category was selected that brought together all other categories into an explanatory whole and a matrix of meaning was constructed to schematically present the relationships between concepts. The scheme was completed by returning it to the interviewees who verified that the researchers’ interpretations corresponded to what they wanted to express [16].

## 3. Results

Fourteen physicians were included with an age range between 24 and 60 years old; six had some clinical specialty. The years of work experience ranged from 1 to 25 years. Eleven reported a family history of cancer, one reported personal history of lymphoma, and only two did not report personal or family history of the disease (Table 1).

### 3.1. Etiology of Cancer

The etiology of cancer is understood by all participants as the interaction between genetic, environmental, and lifestyle factors. In addition, the category of psychogenesis of cancer emerged which refers to the narratives of a group of physicians with differences in their training, experience, and beliefs for whom emotional and “mental” factors influence the development of the disease. For the interviewees, there were no substantial differences between emotions and the mind; that is, they were notions taken as equivalent. In the psychogenic category, four mechanisms were described. The first refers to emotional resonance to describe the fact that some physicians attribute to certain emotions the property of directly transforming matter, physical health. The second mechanism involves the neurological, endocrine, and immune systems; in their opinion, chronically repressing or experiencing certain emotions could stimulate the production of neurotransmitters, affect the immune response against tumor cells, and lead to the development of the disease.

“When emotions are not properly managed, all that mental part can have an impact on something physical, and that physical part can trigger something cellular and subsequently trigger some type of disease, in this case, cancer” (E008).

“Completely convinced that everything we work on in the mind affects disease or health. A positive mind, a calm mind, a healthy mind, is probably attached to a healthy body. A mind that holds grudges, that holds anger, hatred, develops diseases and cancer is one of them” (E004).

“Our thoughts have an impact on our entire being. Positive thoughts or negative thoughts will bring a load of stress or peace to us; this will influence our endocrine and immune system and this obviously has an impact on the other systems. I am completely sure that they have a great impact on the development of the disease” (E005).

The third mechanism links certain negative attitudes (pessimism, hopelessness, anxiety) with gene methylation, the appearance of mutations, and subsequent development of cancer. It is important to note that for some interviewees, attitudes were considered a matter located on the mental plane. The fourth mechanism attributes omnipotent qualities to the mind and, based on models of other cases such as the placebo effect or phantom limbs syndrome, attributes it the ability to create cancer.

“It is believed, although this is not proven, that probably attitude, there is something called epigenetics, may eventually have something to do with these methylations and the appearance of mutations. This could be a point of current research and finally reach that conclusion. I think there is still a long way to go in that scenario, but I think there are some hypotheses about it” (E003).

“For me, the mind can do everything. And why? Well, my most basic explanation is, for example, phantom limbs. One has a limb amputated and still feels it. The mind can do everything” (E006).

### 3.2. Conceptualization of Positive Thinking

Positive thinking emerges as a key aspect of coping with the disease. This expression is used by participants to account for, on the one hand, the development of a mental training discipline that leads patients to think positively and practice visualization exercises that help people focus on their health and overcome the disease. On the other hand, the expression is used to refer to optimism, the expectation that everything related to their illness and treatment will work well, and the belief that this style of thinking is effectively related to better health outcomes.

“I believe that people who are positive about anything trust, believe, and do not stress. Let’s see, I think it will go well for me tomorrow! so I sleep peacefully today because I think it will go very well for me tomorrow, right? So I rest and tomorrow I wake up as if the world will work well. And if things don’t work out so well, I handle them without much anguish… That would be being positive, trusting that things will work out well. If I’m not confident about that, if I’m thinking: “no, that will surely go wrong”, “no, that will surely get tangled up”, then I’m getting anxious ahead of time, I’m releasing cortisol ahead of time, I’m releasing adrenaline and noradrenaline ahead of time, I’m predisposing my body to negative situations, stressing it unnecessarily. So that eventually makes me not recover properly because I’m stressing myself” (E004).

“Deepak Chopra’s visualization therapies are something that I think can help the person understand their illness more, have more fighting spirit, a more positive mind and obviously without neglecting the other treatments can improve” (E010).

This category led to the construction of a patient typology in which positive and negative patients were differentiated. A positive patient is described as having qualities such as optimism, an attitude focused on the benefits of the situation, good adherence, and a good response to treatment. On the other hand, a negative patient is described as one who feels overwhelmed by the disease, has feelings of frustration, has a bad attitude, is pessimistic about their future, and does not respond well to treatment. It is important to mention that in this classification, there is no allusion in the speeches to the individual experiences of the patients and to the different stages of the adaptation process involved in receiving an oncological diagnosis.

“Patients who are positive about their illness are those who face the illness thinking that they can do well and that they have the possibility of curing their illness or if not of curing it, at least of having an adequate treatment for the illness and generally these patients do better. Negative patients are those who are told they have cancer and automatically say they will die” (E007).

“Surely we put a person on one side, with the same type of cancer, at the same stage and another person on the other side, with the same type of cancer, at the same stage, surely the person with a better attitude, with a better emotional state, surely can have a much more positive response to treatment than the other” (E001).

### 3.3. Treatment of Cancer

Treatment is also conceived as a multifactorial process involving biomedical work (surgical, chemotherapy and/or radiotherapy), social work, and psychiatric dimensions. In addition to this and consistent with the etiological factors of cancer, psychological aspects emerged as an important factor in the treatment of patients, and mechanisms were enunciated that include fighting spirit, emotion management, psychoneuroimmunological and psychoneuroendocrinological mechanisms. The fighting spirit is characterized by an optimistic view of the disease, a determination to fight against cancer and not allow it the altering of everyday life. This affects treatment response in two ways: improving adherence to treatment or affecting therapeutic efficacy itself.

“Well, in addition to the traditional treatments given by specialists in the field, such as surgical, chemo, and radiotherapy, I think it is extremely important to treat the psyche and soul of these people. That person has to work through their sadness, their resentments, their anger in order to move forward and also give them a positive mindset, that is what I was telling you earlier. You have to give them hope, you have to give them options, you have to let them know that they can move forward so that they can develop I don’t know what, I don’t know what it will be that one takes from their body in order to face the illness and come out ahead” (E004).

“Obviously a patient who enters already demotivated, a patient who enters stressed out with negative attitudes is probably a patient who from a pharmacological point of view may abandon management, may abandon treatment and from an emotional point of view we had already talked about how this affects our different systems (immune, endocrine)” (E005).

In relation to the immune response, the interviewees mentioned that positive thinking could act by preventing the release of proinflammatory substances produced under conditions of stress. Regarding psychoneuroimmunological and psychoneuroendocrinological aspects, some doctors observed a relationship between mood states, neurotransmitter production, immune system affectation and therapeutic outcomes.

“It is seen that when the body is subjected to a situation of stress, whether physical or psychological stress, certain proinflammatory substances are released. So when a patient who has, let’s say, positive thoughts and is not in a stressful situation, has less chance of releasing those substances, so in treatment they could do better” (E007).

“We as doctors provide the part, so to speak, the labor part, chemo, radiotherapy, surgery. But part of the recovery and part of all that also depends on the patient’s mood state, how they feel, because this obviously alters the immune response, alters recovery, healing, the hormonal axis, the adrenergic axis” (E001).

After establishing relationships between categories and subcategories, it becomes clear that the ideology of positive thinking has managed to permeate medical discourse and the representations that are formed about the etiology and treatment of cancer. Although cancer is conceived as a multicausal phenomenon and therefore requiring a holistic and multidisciplinary treatment, doctors place the mind, emotions, attitude, and positive thinking as determinants of the origin of the disease and response to therapy. It is important to emphasize that, although disciplinarily (that is, in the field of psychology) these categories are not unitary, physicians understand them as equivalent and embrace them under the broad category of “mental”. To argue this link, they resort to two strategies: (i) a sophisticated and specialized discourse that involves relationships among thoughts, genetics, and the neurological, immune, and endocrine system, and (ii) a mystical and less rational discourse that emphasizes the omnipotence of the mind and thoughts. In no case is the idea of positive thinking rejected or disagreement expressed with this style of thinking (Figure 1).

## 4. Discussion

Positive thinking emerged as an explanatory category in the etiology and treatment of cancer, involving mechanisms that include the immune, endocrine, and neurological systems. To understand the origin of these ideas, we must refer to the studies of Hans Selye, who linked stress to health effects. Selye conducted experiments with rats, subjecting them to stressful situations such as extreme cold or continuous running on rotating wheels, and found that this resulted in adrenal hyperactivity, lymphatic atrophy, and peptic ulcers. He concluded that the body responds to stress through various mechanisms involving the hypothalamic—pituitary–adrenal axis [18]. In a subsequent experiment, Rober Ader fed rats a combination of water, saccharin, and a drug that induced nausea. Later, the rats were only fed water mixed with saccharin; however, even without the drug, most of them died. The researcher concluded that the rats’ immune system was affected, through classical conditioning, by the stress of drinking water associated with nausea [19]. This study was foundational in the field of psychoneuroimmunology, a discipline that studies the interactions between the nervous system and the immune system. Other authors examined these findings and equated stress with negative and positive thoughts as a way to counteract it. In this regard, oncologist Carl Simonton [20] argued that cancer occurs due to a weak immune system, and given the connections between the immune and the nervous system suggested by psychoneuroimmunology, thoughts play an important role in the etiology and treatment of the disease. These ideas align with the testimonies of the physicians involved in this research; however, different studies have discredited them. In fact, the American Cancer Society [21], which publishes information targeted at patients, includes an explicit text fragment to reject these ideas: “Studies have shown that keeping a positive attitude does not change the course of a person’s cancer. Trying to keep a positive attitude does not lead to a longer life and can cause some people to feel guilty when they can’t “stay positive.” This only adds to their burden”.

This way of understanding the disease is not harmless. As mentioned before, a characteristic of positive thinking is that individuals are expected to assume some responsibility for their recovery and to seek psychological factors (hatred, resentment, negative thoughts, etc.) rather than biological or sociocultural factors that contributed to their illness. This places a significant burden on patients as promoting self-responsibility opens the door to blame [22]. The ideas of responsibility and blame are strongly intertwined in our language and everyday life. Blaming patients for their illness results in the loss of the sick role. Simultaneously, it gives way to social moralization as individuals become subject to censure for personal failings that “caused” their condition or their inability to influence therapeutic success. Furthermore, it gives rise to medical paternalism, as doctors assume authority over a patient’s body that they have failed to protect on their own [23]. It has also been reported that patients feel shame and despair when they fail to “prevent” the progression of cancer [15].

The reference made by some of the participating physicians in this study to self-help literature and authors such as physician Deepak Chopra is important because self-help literature constitutes one of the main devices through which the ideology of positive thinking is mobilized [7]. In this regard, Vanina Papalini [24] refers to the concept of bibliotherapy or the use of books as a therapeutic tool. The author mentions that these texts are used as a complement to specific treatment or as a therapy in itself. In the former case, books are integrated without replacing biomedical therapy, while in the latter, books promote a set of practices that aim to heal the affected individual. The historical precedent of bibliotherapy lies in the therapeutic function of words used in performative acts such as prayer or magical words, which also provide comfort to the patient’s spirit, as the disease is understood because of a life where the repression of emotions, hatred, or resentment play central roles, and they also promote prevention by conveying recommendations for a healthy life. However, in the specific case of self-help literature in the field of cancer, Chopra himself argues that cancer can be cured by visualizing wellness, and Louise L. Hay [25] asserts that cancer can be cured by abandoning the mental model that has promoted the disease. These ideas, while proposed by some physicians as a complement to conventional treatment, are harmful to patients because books act as individualizing devices by locating the etiology and therapy of cancer in the personal domain; they speak, as pointed out by Beck and Beck-Gernsheim [26], “of seeking biographical solutions to systemic contradictions”.

In the discourse of the physicians participating in this research, the social and political determinants of cancer do not appear. In contrast, there is a particular emphasis on individual responsibility, both in terms of etiology and therapeutic effectiveness. This understanding of disease is characteristic of the ideology of positive thinking, shares points of convergence with neoliberal ideology, and hinders effective measures for disease prevention as it depoliticizes the socio-structural causes of cancer and supports the status quo of the dominant economic system that has led to poor health. Waitzkin [27], in a Marxist analysis of health systems in advanced capitalist societies, exemplifies this problem with acute myocardial infarction. The author argues that this condition is closely linked to the social stress experienced by individuals in their jobs or economic difficulties. However, the emphasis in interventions has been placed on individual stress and techniques such as meditation focused on reducing perceived stress levels. Thus, little attention has been paid to reducing social stress as a way to prevent heart attacks, ignoring the structural sources of stress stemming from the modern industrial system or assuming that the necessary changes are so structural that they are impractical to implement. The Marxist viewpoint questions whether significant improvements in the health system can occur without fundamental changes in the broader social order. In this case, as in the testimonials of the participants in this study, the focus on the disease has been on individual changes, and therefore, without bringing visibility to the structural causes, the structural causes of cancer are not affected.

Another important finding of this study is the reference made by the physicians to the mind as if it were a central reference point that commands, orders, and conditions the body and its characteristics. Despite some interviewees mentioning the need for integration between the mind and body, they articulate them as separate spheres that need to be balanced, thus justifying the intervention in thoughts to positively impact the body. This is interesting because epistemologically, the notion of the mind has been used to contrast it with the body; this notion relates to a very old philosophical problem: the mind–body dualism elaborated by Descartes. This finding coincides with a study on social representations of the body in medical students in Medellín, where the authors found that the body–mind division is also strongly ingrained in the conceptions of the interviewed students [28]. The convergence of these results serves as a good example of what Ian Hacking [29] and Lorraine Daston [30] understand as applied metaphysics; that is, how old philosophical ideas “seep” into everyday life or professional life, determining and guiding people’s actions. In this sense, healthcare professionals need to transition towards holistic models of patient care that transcend the artificial separation of the “physical” and the “mental,” allowing for a more humanistic practice of medicine while maximizing the likelihood of treatment success [31].

The terms used by physicians to refer to the relationship between thoughts, attitude, mind, and cancer provide the scientific legitimacy that these ideas require to become performative [32]. This symbolic capital of physicians influences the authority of their discourse over patients, who tend to passively accept what doctors say and do not feel qualified to discuss or criticize their recommendations [12]. In addition, in analyses of the physician–patient relationship from a political economy perspective, it is argued that despite contemporary Western societies having more access to information sources on the internet and scientific literature about health, clinical decisions are not solely based on knowing the figures or algorithms but on an “intuitive” competence that is acquired over time. In this sense, there still remains an aura of esotericism surrounding medical knowledge, resulting in patients having limited opportunities to judge the quality of the recommendations they receive and influence their therapeutic course. In this context, patients are forced to rely on physicians to make decisions about their health. This is known as the competence gap and serves to support the dominant position of the medical profession and the dependence of patients. In the specific case of cancer, the patient’s need to depend on medical authority is greater because patients face problems with unclear medical solutions, mixed with social difficulties and moral dilemmas, causing them to prefer that doctors take control of the situation that scares and causes anxiety [12]. Therefore, it is advisable for physicians to avoid transferring these ideas to patients as they may perceive them as scientific facts, resulting in an additional burden to their illness and even leading them to adopt pseudoscientific practices that deviate from conventional biomedical treatments.

It is important to consider the limitations of this study. The fact that oncologists were not included to provide their testimonies on the social representations of cancer is a significant limitation. Note that the Ministry of Health does not have a registry of specialists, and according to references from the Colombian Association of Scientific Societies, there are only about 250 oncology specialists in the entire country, at best [33]. This situation made it challenging to contact oncologists and involve them in the study. Similarly, it is important to emphasize that this research reflects the social representations of physicians and, although these representations may influence their clinical practice, they should not be confused with it.

## 5. Conclusions

In conclusion, the ideology of positive thinking is a powerful narrative of self-realization and personal growth that attributes the cause of all problems to psychological and individual deficiencies while legitimizing the absence of social determinants. These ideas have managed to permeate the field of illness and even scientific education, as evident in the discourse of physicians. This research accounts for this phenomenon in relation to cancer; however, there are indications that the ideology of positive thinking may be related to other health issues as well. It has generated new subjectivities, altered the ways in which life is interpreted, and shaped beliefs about how life should be lived. This work serves as a starting point for researchers to delve deeper into identifying the social agents who find the dissemination of these ideas useful, the political and economic interests it serves, and opens the door for further investigation on this topic.

## Figures and Tables

**Figure 1 behavsci-13-00866-f001:**
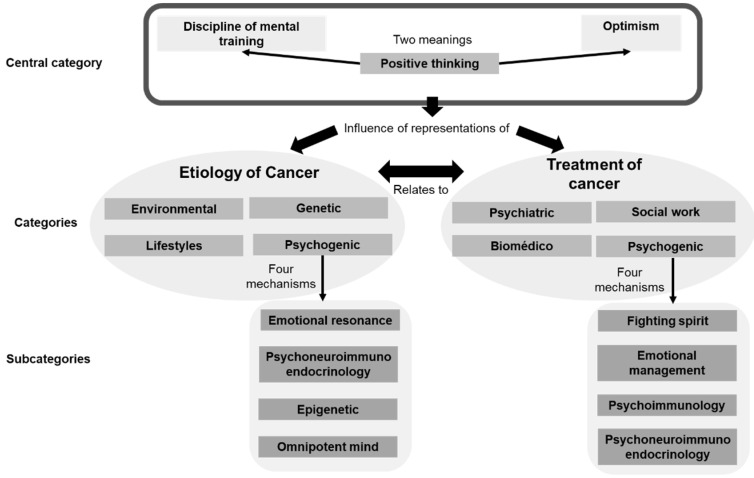
Significance matrix of cancer representations among physicians.

**Table 1 behavsci-13-00866-t001:** Description of the characteristics of the physicians.

Code	Age (Years)	Sex	Clinical Specialty	Years of Work Experience	Religion	Personal or Family History of Cancer
E001	24	Male	General practitioner	<1 year	Without religion	Family (cousin), breast cancer
E002	50	Male	Urgentologist	25 years	Catholic	Personal, lymphoma
E003	31	Male	Urologist	7 years	Catholic	Family (father)
E004	64	Female	Otolaryngologist	40 years	Catholic	Family (mother), gastric cancer
E005	27	Male	General practitioner	2.5 years	Christianity	Family (aunt), breast cancer
E006	36	Female	Ophthalmologist	13 years	Catholic	Nobody
E007	39	Male	Urologist	15 years	Catholic	Family (aunt), stomach cancer
E008	26	Male	General practitioner	2 years	Without religion	Family (nephew), leukemia
E009	26	Female	General practitioner	<1 year	Catholic	Nobody
E010	25	Male	General practitioner	2 years	Catholic	Family (grandmother), pharyngeal cancer
E011	42	Female	Pediatrician	18 years	Catholic	Family (grandmother), breast cancer
E012	60	Male	General practitioner	28 years	Catholic	Family (mother), colon cancer
E013	24	Male	General practitioner	<1 year	Catholic	Family (aunt), thyroid cancer
E014	26	Female	General practitioner	4 years	Catholic	Family (grandmother), pancreatic cancer

## Data Availability

The data were not deposited in a public repository. Anonymized data are available upon reasonable request.

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
