# Peer review of "Influence of Positive Thinking Ideology on Physician Representations of Cancer"

_behavsci, 2023, doi:10.3390/bs13100866_

Round 1
Reviewer 1 Report
Lines 105-108 need more context regarding the ontoepistemiological justification for usage of grounded theory
It should be denoted the beginning part of the introduction or the abstract the authors' bad definition of physicians. Given the table represents individuals across the spectrum of preventative care.
Lines 120-131- The authors need to explain their purpose of employing semi-structured interviews outside just their application of grounded theory. How do they style of interview contribute qualitatively not just theoretically.
The authors are discussing cancer yet there is no a brief exposition or definition of what cancer is characterized in the scholarly literature of medicine. This will be helpful to the readers as there are many forms of cancer biologically.
Author Response
Thank you for your guidance in reviewing our submission. The manuscript has been revised and the reviewers’ comments have been addressed below. We are thankful to the reviewers for their valuable suggestions for improving the manuscript, and we hope it is now acceptable for publication. Responses to each reviewer are included in yellow highlights within the manuscript.
Thank you for your consideration.
Sincerely,
The authors

Reviewer 2 Report
The aim of this study is to investigate the influence of positive thinking ideology on cancer representations among physicians in Medellín. Using a qualitative approach based on Corbin and Strauss’s grounded theory, the study includes fourteen physicians selected for maximum variation in education, years of study, and personal and family history of cancer.
Q1. Can the authors elaborate on the significance of the psycho-genesis category and how it was identified in the narratives of the physicians? Were there commonalities or differences among physicians with varying training, experience, and beliefs?
Q2. The text mentions four mechanisms within the psychogenic category. How did these mechanisms emerge during the authors' research, and were they consistent across all interviewed physicians?
Q3. In the context of the third mechanism linking attitudes with gene methylation and the development of cancer, could the authors provide examples of attitudes that were considered relevant, and how were these attitudes perceived as being located on the mental plane?
Q4. The text refers to a belief that emotions, when not properly managed, can impact physical health. How did physicians express this connection between emotions, mental well-being, and physical health during the interviews, according to the authors?
Q5. Were there variations in the understanding of the mind-body connection among physicians with different religious backgrounds or those who identified as having no religion, as observed by the authors?
Q6. The fourth mechanism attributes omnipotent qualities to the mind. Could the authors elaborate on how physicians conceptualized the mind's ability to create cancer, drawing parallels with phenomena like the placebo effect or phantom limbs syndrome?
Q7. Did the authors observe any patterns or variations in the physicians' perspectives based on their clinical specialties? For example, did oncologists have different views compared to general practitioners or specialists in other fields, according to the authors' findings?
Q8. How might the findings of the authors' research inform discussions within the medical community about the multifaceted nature of cancer etiology and the potential implications for patient care and support?
The manuscript meets basic standards of English proficiency. The language is clear and generally understandable. However, there is room for improvement to enhance the overall quality of expression. Some sentences could be refined for clarity and conciseness.
Author Response

(The authors gave the same response as above.)
